# Multiple acyl-CoA dehydrogenase deficiency kills *Mycobacterium tuberculosis* in vitro and during infection

Tiago Beites [1], Robert S. Jansen [2,3], Ruojun Wang [1,4], Adrian Jinich[2], Kyu Y. Rhee [1,2], Dirk Schnappinger[1] & Sabine Ehrt [1✉]

The human pathogen *Mycobacterium tuberculosis* depends on host fatty acids as a carbon source. However, fatty acid β-oxidation is mediated by redundant enzymes, which hampers the development of antitubercular drugs targeting this pathway. Here, we show that *rv0338c*, which we refer to as *etfD*, encodes a membrane oxidoreductase essential for β-oxidation in *M. tuberculosis*. An *etfD* deletion mutant is incapable of growing on fatty acids or cholesterol, with long-chain fatty acids being bactericidal, and fails to grow and survive in mice. Analysis of the mutant's metabolome reveals a block in β-oxidation at the step catalyzed by acyl-CoA dehydrogenases (ACADs), which in other organisms are functionally dependent on an electron transfer flavoprotein (ETF) and its cognate oxidoreductase. We use immunoprecipitation to show that *M. tuberculosis* EtfD interacts with FixA (EtfB), a protein that is homologous to the human ETF subunit β and is encoded in an operon with *fixB*, encoding a homologue of human ETF subunit α. We thus refer to FixA and FixB as EtfB and EtfA, respectively. Our results indicate that EtfBA and EtfD (which is not homologous to human EtfD) function as the ETF and oxidoreductase for β-oxidation in *M. tuberculosis* and support this pathway as a potential target for tuberculosis drug development.

[1] Department of Microbiology and Immunology, Weill Cornell Medical College, New York, NY 10065, USA. [2] Division of Infectious Diseases, Department of Medicine, Weill Cornell Medical College, New York, NY 10065, USA. [3] Present address: Department of Microbiology, Radboud University, 6525 AJ Nijmegen, The Netherlands. [4] Present address: Department of Molecular Biology, Princeton University, Princeton, NJ 08540, USA. ✉email: sae2004@med.cornell.edu

Maintenance of an energized membrane is essential for *Mycobacterium tuberculosis* (Mtb) to grow and survive periods of non-replicating persistence[1]. This need has driven tuberculosis (TB) drug development efforts towards Mtb's energy metabolism. These efforts are supported by the first anti-TB drug approved in over 40 years—the ATP synthase inhibitor bedaquiline[2].

Mtb's energy related pathways exhibit varying degrees of vulnerability to inhibition. Uptake of its main carbon sources in vivo is performed by specialized transporters: the multi-subunit Mce1 complex transports fatty acids[3] and the Mce4 complex facilitates the uptake of cholesterol[4]. Inactivation of Mce1 can reduce intracellular growth[5], although it has also been reported to cause hypervirulence[6], while Mce4 was conclusively shown to be essential for survival during the chronic phase of infection[4]. LucA, which acts as a regulator of both Mce1 and Mce4, is also required for wild type levels of Mtb virulence[3], further supporting that inhibiting the ability to import host lipids affects Mtb's pathogenicity.

Cholesterol degradation yields multiple products, including acetyl-CoA, propionyl-CoA, pyruvate, and likely succinyl-CoA that can be used for energy generation or lipid biosynthesis[7–11]. Deletion of genes encoding cholesterol oxidation enzymes attenuated growth[12] and caused survival defects of Mtb in mice[13,14]. A screen against Mtb residing in the phagosomes of macrophages identified several compounds that target cholesterol metabolism[15]. In contrast, fatty acids are degraded solely through β-oxidation. Mtb's genome encodes multiple enzymes for each step of β-oxidation, including 34 putative acyl-CoA ligases, 35 putative acyl-CoA dehydrogenases, 22 putative enoyl-CoA dehydratase, five putative β-hydroxyacyl-CoA dehydrogenase, and six putative thiolases[16]. Some of these enzymes are necessary for infection, but they also have been shown to play roles in other pathways, such as complex lipid biosynthesis[17] and cholesterol degradation[18]. Due to the apparent redundancy of Mtb's fatty acid β-oxidation machinery, this metabolic pathway has thus been presumed to be invulnerable to chemical inhibition.

In the present study, we define an enzyme complex that has previously not been recognized as required for fatty acid degradation in Mtb. It consists of an electron transfer flavoprotein composed by two subunits—FixA (Rv3029c) and FixB (Rv3028c)—and a membrane oxidoreductase (Rv0338c), which we propose to re-name as EtfB$_{Mtb}$, EtfA$_{Mtb}$, and EtfD$_{Mtb}$ based on their human counterparts. Deletion of Mtb's EtfD causes multiple acyl-CoA dehydrogenase deficiency, which prevents utilization of fatty acids as carbon sources and can kill Mtb in vitro and during mouse infection.

## Results

### A possible role for EtfD$_{Mtb}$ in Mtb's energy metabolism.
EtfD$_{Mtb}$ (Rv0338) is a membrane protein of unknown function predicted to be essential for growth of Mtb on agar plates[19,20]. To experimentally determine its topology, we fused *E. coli* alkaline phosphatase PhoA to EtfD at specific residues of the predicted transmembrane helices. Transport of PhoA outside of the cytoplasm enables reactivity with the chromogenic substrate 5-bromo-4-chloro-3-indolyl phosphate p-toluidine (BCIP). Based on this assay, the soluble portion of EtfD$_{Mtb}$ faces the cytoplasm (Supplementary Fig. 1a), and this topology agrees with the prediction generated by the MEMSAT3 algorithm[21] (Supplementary Fig. 1b).

Next, we performed in silico analysis, which placed EtfD$_{Mtb}$ into the cluster of orthologue groups 0247 (COG0247) composed of Fe-S oxidoreductases involved in energy production and conversion (Fig. 1a). COG0247 includes a variety of enzymes, including lactate dehydrogenase and the methanogenic-related heterodisulfide reductase; however, most proteins in COG0247 remain uncharacterized and are of unknown function. EtfD$_{Mtb}$ is predicted to be a chimeric enzyme assembled from four domains: an N-terminal domain, which is similar to the gamma subunit of nitrate reductases and putatively binds a cytochrome b; a central 4Fe-4S di-cluster domain similar to succinate dehydrogenases; and two C-terminal cysteine-rich domains (CCG), which are often found in heterodisulfide reductases (Fig. 1b). This analysis led us to hypothesize that EtfD is a component of Mtb's energy metabolism.

### EtfD$_{Mtb}$ is linked to fatty acid metabolism and it is essential in vivo.
To analyze the impact of EtfD$_{Mtb}$ depletion on Mtb's growth in vitro, we generated a TetOFF strain, in which EtfD levels are controlled by anhydrotetracycline (ATC)-inducible proteolysis[22]. Depletion of EtfD$_{Mtb}$ inhibited Mtb's growth in regular medium supplemented with oleic acid, albumin, dextrose and catalase (OADC), which is consistent with its predicted essentiality (Fig. 1c, d). Curiously, when the same medium was supplemented with a fatty acid free enrichment (albumin, dextrose and sodium chloride, ADN), depletion of EtfD$_{Mtb}$ did not impact growth to the same extent (Fig. 1c, d). This is similar to the fatty acid sensitive phenotype observed in response to inactivation of Mtb's type II NADH dehydrogenases[23], which allowed the genetic deletion of Ndh-2 by growth in a fatty acid free medium. We applied the same strategy to etfD$_{Mtb}$ and generated a deletion strain (ΔetfD) (Supplementary Fig. 2a). The ΔetfD strain was confirmed by whole genome sequencing (WGS) to not contain additional polymorphisms known to affect growth (Supplementary Table 1). This knockout strain also phenocopied the fatty acid sensitivity observed with the knockdown mutant (Supplementary Fig. 2b, c).

We next sought to investigate the importance of etfD$_{Mtb}$ for pathogenesis in an aerosol model of TB infection in mice. After aerosol infection, ΔetfD was unable to grow in mouse lungs and declined in viability from day 14 onwards (Fig. 2a and Supplementary Fig. 3a). In agreement with the CFU data, gross lung pathology showed no lesions in the mice infected with ΔetfD (Fig. 2b and Supplementary Fig. 3b). The attenuation was even more pronounced in spleens, where no ΔetfD CFU were recovered at any time point (Fig. 2a and Supplementary Fig. 3a). All phenotypes were rescued by reintroducing an intact copy of etfD$_{Mtb}$.

### Mtb requires EtfD$_{Mtb}$ to consume fatty acids and cholesterol as a carbon source and to prevent toxicity of long-chain fatty acids.
The fatty acid sensitivity of ΔetfD suggested that this protein is, directly or indirectly, required for fatty acid metabolism. To test this further, we grew strains in media with different single carbon sources. ΔetfD was able to grow in both glycolytic (glycerol) and gluconeogenic (acetic acid and propionic acid) carbon sources, although at a slower rate than the wild type (Fig. 3a). However, longer chain fatty acids (butyric acid, palmitic acid and oleic acid) did not support detectable growth of ΔetfD (Fig. 3b). Fatty acids with four carbons or more in length require functional β-oxidation to be utilized, hence these results indicated that ΔetfD might display a defect in β-oxidation. Mtb oxidizes the cholesterol side-chain through β-oxidation. Enzymes involved in cholesterol side-chain degradation are essential for Mtb to grow with cholesterol as single carbon source[7,12]. Similarly, ΔetfD was not able to grow with cholesterol as a single carbon source (Fig. 3c).

To distinguish between two possible interpretations, namely 1) ΔetfD simply failed to use fatty acids as carbon source and 2) ΔetfD was intoxicated by fatty acids, we tested if glycerol could

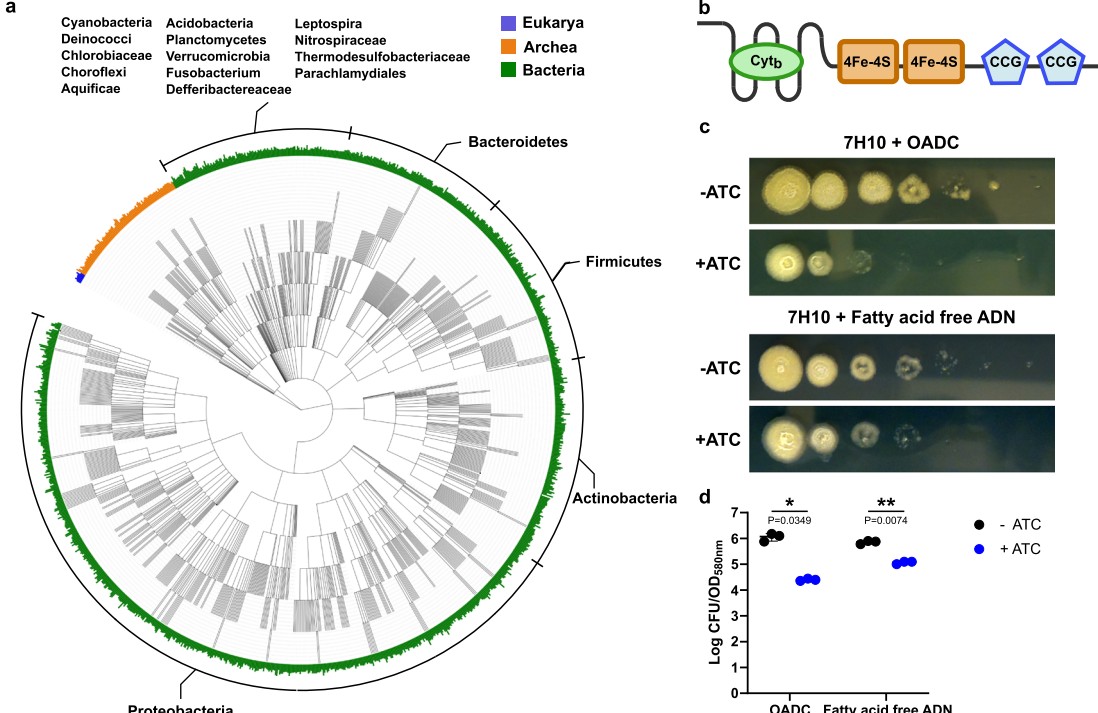

**Fig. 1 EtfD is a membrane protein possibly involved in Mtb's energy metabolism. a** Rootless phylogenetic tree with species containing members of COG0247 according to the database of the webtool EggNog. The inner colored ring corresponds to life domains, while the outer ring refers to bacteria phyla. **b** Domain architecture of EtfD based on the algorithms of HHPred and Xtalpred. $Cyt_b$ cytochrome b, CCG cysteine-rich domain. **c** Spot assay on solid media with an EtfD-TetOFF strain, where protein levels are controlled by anhydrotetracycline (ATC). Serial dilutions ($10^6$ down to 1 bacteria) were incubated for 14 days. These results are representative of three independent experiments. **d** Quantification of spot assays (**c**). This quantification aggregates the data from the three independent experiments. To quantify the spot assays, we picked the first dilution where isolated CFUs could be counted and divided it by the optical density (OD) of the corresponding bacterial suspension. The discrepancy between the OD of the bacterial suspension and the corresponding CFUs, which is due to sensitivity to lipids in the solid media, is reflected by this ratio. Data are averages with individual data points depicted. Error bars correspond to standard deviation. Statistical significance was assessed through paired two-tailed $t$-test. *$P < 0.05$; **$P < 0.01$.

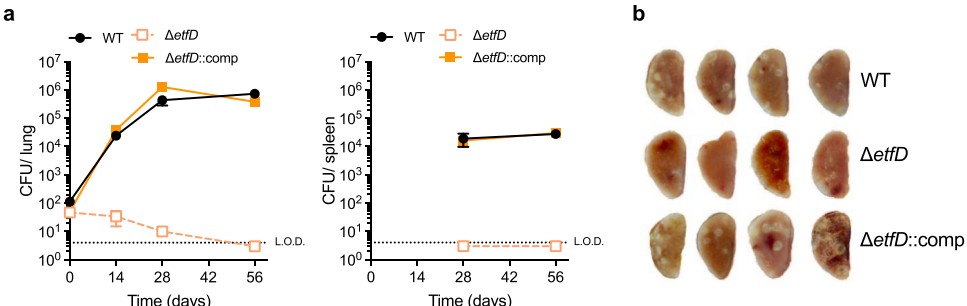

**Fig. 2 EtfD is essential for growth and survival in vivo. a** Growth and persistence of wild type Mtb, Δ*etfD* and the complemented mutant in mouse lungs and spleens. Data are CFU averages from four mice per time point and are representative of two independent experiments. Error bars correspond to standard deviation. "Comp" stands for complemented. L.O.D. stands for limit of detection. **b** Gross pathology of lungs infected with wild-type Mtb, Δ*etfD* and the complemented mutant at day 56.

rescue the impaired growth of Δ*etfD* with fatty acids. Glycerol was able to restore growth in medium with butyric acid as carbon source, but it was not able to restore growth with long-chain fatty acids. Cholesterol partially inhibited growth in this mixed carbon source setup (Supplementary Fig. 4). This argued for a toxic effect and led us to evaluate the impact of fatty acids and cholesterol on the viability of Δ*etfD*. We found that butyric acid and cholesterol were not toxic, while long-chain fatty acids were bactericidal to Δ*etfD* (Fig. 3d). Glycerol did not rescue the bactericidal effect of long-chain fatty acids, and partially rescued growth with cholesterol in the medium (Fig. 3d and Supplementary Fig. 4).

**Mtb acyl-CoA dehydrogenase activity requires EtfD$_{Mtb}$.** We applied metabolomics to understand why the consumption of fatty acids is prevented in the absence of EtfD$_{Mtb}$. We focused these studies on $^{13}C_4$-labeled butyric acid to isolate the impact of *etfD* disruption on fatty acid consumption from potentially additional confounding effects associated with the toxicity of longer chain fatty acids. Inspection of labelled metabolites in central carbon metabolism confirmed the presence of butyric acid in WT, Δ*etfD*, and the complemented mutant (Fig. 4 and Supplementary Fig. 5). All three strains were thus able to import $^{13}C_4$-labelled butyric acid.

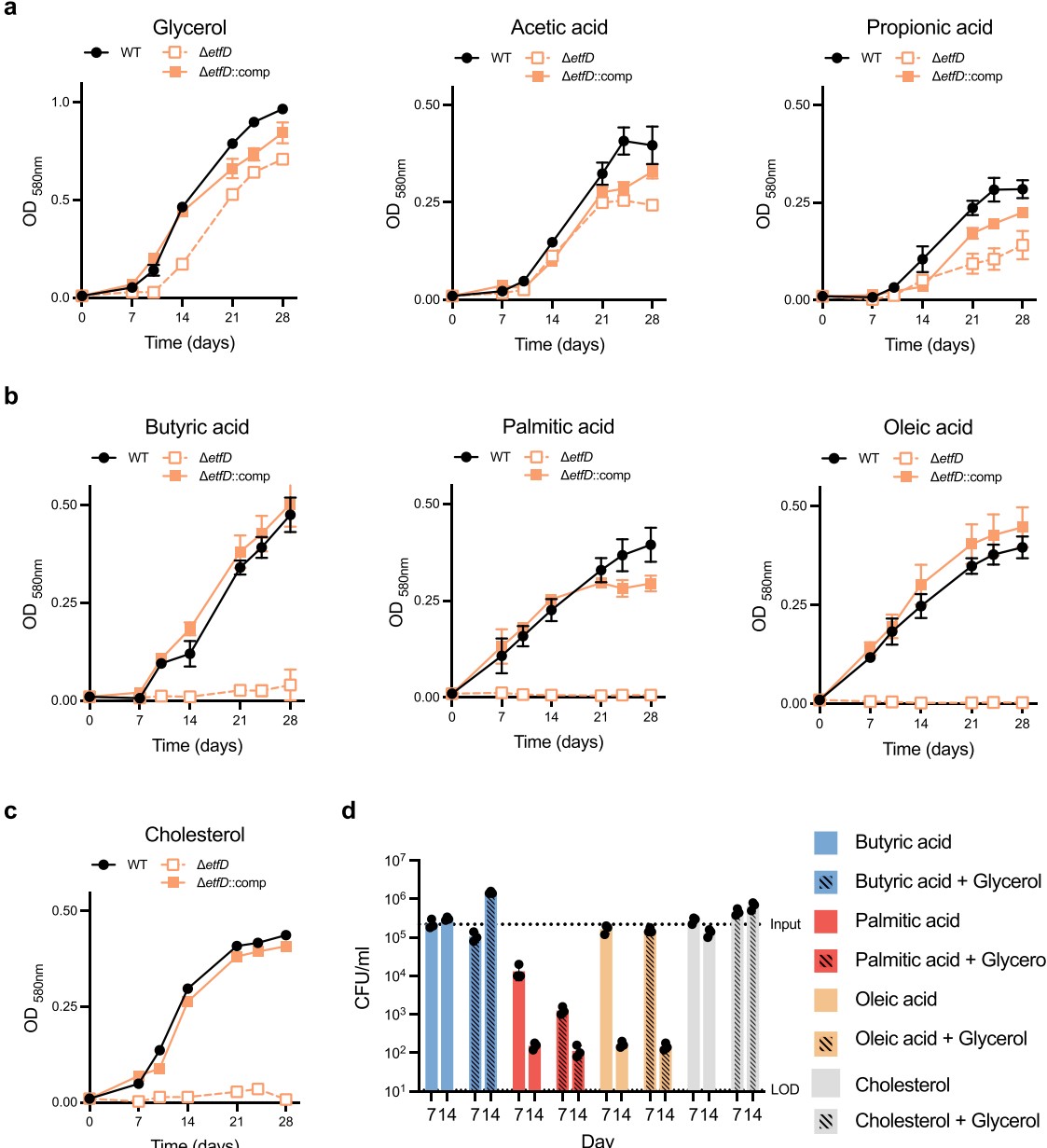

**Fig. 3 EtfD is necessary for the utilization of fatty acids that require β-oxidation. a–c** Strains were grown in media with single carbon sources that **a** do not require and **b** require β-oxidation for catabolism, as well as cholesterol (**c**). Carbon sources were used at the following final concentrations: glycerol 25 mM, acetic acid 25 mM, propionic acid 2.5 mM, butyric acid 2.5 mM, palmitic acid 250 μM, oleic acid 250 μM, and cholesterol 250 μM. To sustain growth and avoid toxicity palmitic acid and oleic acid were replenished every 3–4 days for the first 14 days of culture. Cholesterol was replenished every to 3 to 4 days for the first 14 days to minimize precipitation. Data are averages of three replicates and are representative of three independent experiments. Error bars correspond to standard deviation. "Comp" stands for complemented. **d** Viability of Δ*etfD* in media with single carbon sources (butyric acid, palmitic acid, and oleic acid), or in mixed carbon sources (fatty acids and glycerol) at the same concentrations used in **a** and **b** was assessed at days 7 and 14. Data are averages of three replicates and are representative of three independent experiments. Error bars correspond to standard deviation. "Comp" stands for complemented. LOD stands for limit of detection.

To be catabolized through β-oxidation, butyric acid needs to be transformed by an acyl-CoA ligase into butyryl-CoA. Strikingly, butyryl-CoA accumulated in Δ*etfD* approximately 45-fold relative to WT and the complemented mutant. The remaining inter-mediates of butyric acid β-oxidation were not detectable in any of the strains, but the pool size of the end-product acetyl-CoA was approximately 2-fold lower in Δ*etfD* than in WT. Labelled TCA cycle intermediates, with the exception of succinyl-CoA, were also partially depleted in Δ*etfD* (Fig. 4 and Supplementary Fig. 5). The metabolomic profile of Δ*etfD*, specifically the accumulation of butyryl-CoA, strongly suggested that inactivation of EtfD

interfered with the function of acyl-CoA dehydrogenases (ACAD), which in turn impaired fatty acid catabolism.

**EtfD$_{Mtb}$ interacts with an electron transfer protein**. We immunoprecipitated EtfD$_{Mtb}$ and identified putative interacting proteins by mass spectrometry. Among the 49 total hits (Sup-plementary Table 2), 41 were located or predicted to be located at the membrane/cell wall, while the remaining eight were cyto-plasmic proteins. To investigate possible links to β-oxidation—a cytoplasmic process—we focused on the cytoplasmic interactors,

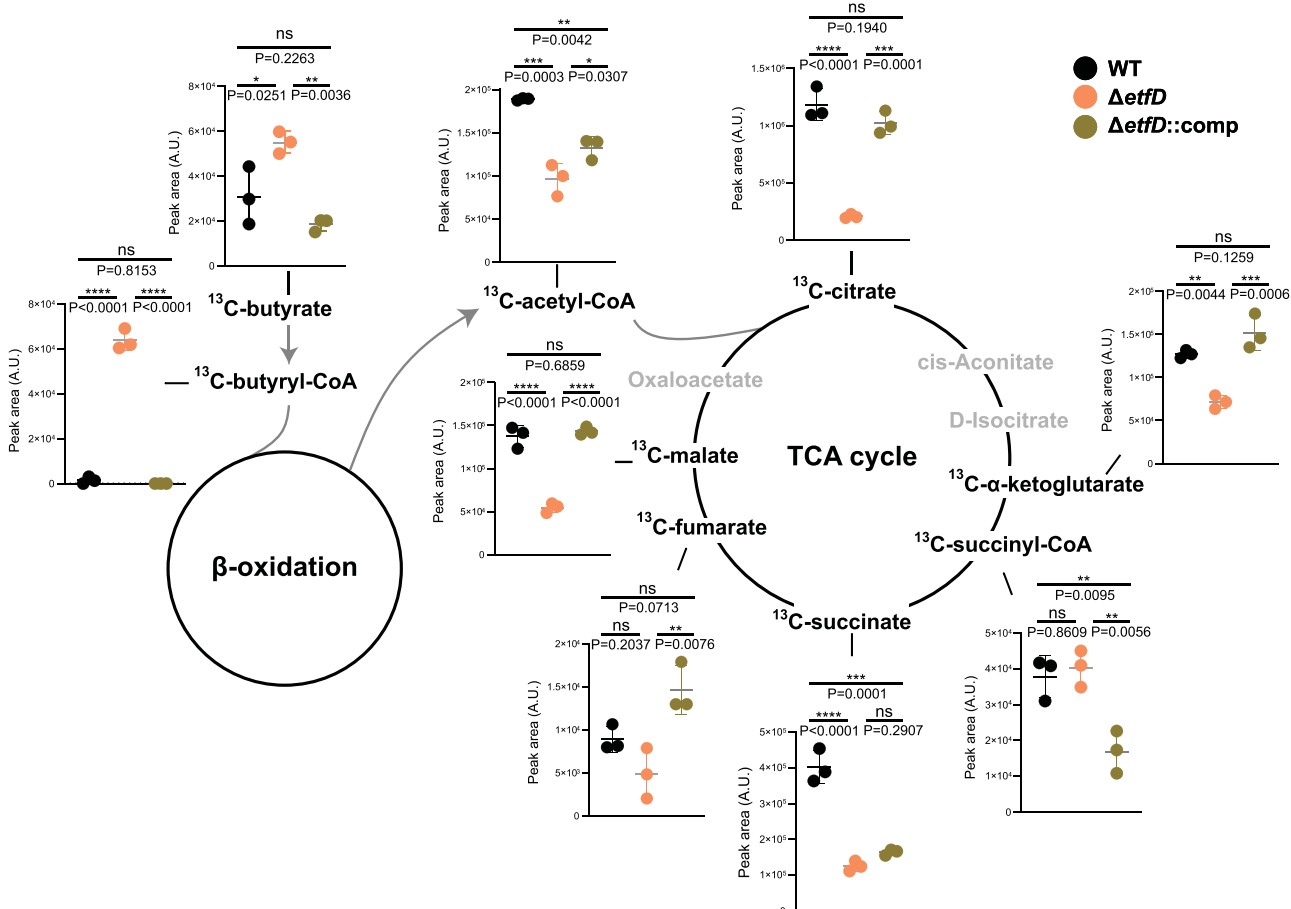

**Fig. 4 Stable isotope tracing reveals a block in β-oxidation at the level of acyl-CoA dehydrogenases.** Strains were grown on filters on top of solid medium permissible to ΔetfD growth for 7 days and then transferred to solid media with $^{13}C_4$-labelled butyric acid (2.5 mM) as single carbon source for 24 hrs. Levels of the indicated $^{13}C$-labeled metabolites (total $^{13}C$ pool, except for $^{13}C_4$-butyrate) were quantified by LC-MS analysis. Data are averages of three replicates and are representative of two independent experiments. Error bars correspond to standard deviation. "Comp" stands for complemented. Statistical significance was assessed by one-way ANOVA followed by post hoc test (Tukey test; GraphPad Prism). *$P < 0.05$; **$P < 0.01$; ***$P < 0.001$; ****$P < 0.0001$. ns not significant. # metabolites with no statistically significant difference between wild-type and ΔetfD in the second independent experiment.

which consisted of two flavoproteins EtfB$_{Mtb}$ and Rv1279, the sigma factor SigA, a putative helicase Rv1179c, an exopolyphosphatase Ppx1, the phthiocerol dimycocerosate (PDIM) biosynthesis enzyme PpsC, the protease ClpP2, and Rv1215c, a protein with putative proteolytic activity (Fig. 5a). We were especially interested in the flavoprotein EtfB$_{Mtb}$, because of its homology (30% identity; 87% coverage) with the beta-subunit of the human electron transfer flavoprotein (ETF). Moreover, etfB$_{Mtb}$ (annotated as fixA—rv3029c) forms an operon with etfA$_{Mtb}$ (annotated as fixB—rv3028c), which shares homology with the alpha subunit of the human ETF (41% identity: 98% coverage). In humans, ETF[24] interacts with a cognate membrane oxidoreductase[25] (EtfD) and both are required to re-oxidize the FAD co-factor of multiple ACADs. This led us to hypothesize that Mtb might display a similar activity. Although EtfD$_{Mtb}$ is not a homolog of the human EtfD, our working model predicted that EtfBA$_{Mtb}$ and EtfD$_{Mtb}$ constitute a complex necessary for the activity of ACADs in Mtb (Fig. 5b).

To further assess a functional connection between EtfD$_{Mtb}$ and EtfBA$_{Mtb}$, we asked if these proteins co-occur across bacterial proteomes. A BLASTp search against a database of 6240 bacterial proteomes (identity cutoff of >30% and coverage cutoff of >75%) identified 469 EtfD$_{Mtb}$, 473 EtfB$_{Mtb}$, and 472 EtfA$_{Mtb}$ homologs, 98% of which occur in actinobacteria, with the spirochete

Leptospira interrogans—the causative agent of leptospirosis—as a notable exception (Fig. 5; and Supplementary Data 1). EtfD, EtfB and EtfA showed a strong co-occurrence ($P$-value <$10^{-10}$), which was suggestive of a functional connection. Curiously, similar to Mtb, L. interrogans is proposed to use host-derived fatty acids as the primary carbon sources during infection[26].

These results support the hypothesis that EtfD$_{Mtb}$ serves as oxidoreductase for EtfBA$_{Mtb}$, which in turn is necessary for the activity of ACADs.

**EtfBA$_{Mtb}$ and EtfD$_{Mtb}$ are required for acyl-CoA dehydrogenase activity.** If EtfBA$_{Mtb}$ and EtfD$_{Mtb}$ participate in the same biochemical pathway then inactivation of EtfBA$_{Mtb}$ should also impair the use of fatty acids and cholesterol as single carbon sources by Mtb. To test this prediction, we first isolated a knockout strain for etfBA in fatty acid free medium (Supplementary Fig. 6) and confirmed its genetic identity through WGS (Supplementary Table 1). We then grew wild type, ΔetfBA and complemented ΔetfBA in media with different carbon sources. ΔetfBA was able to grow with glycerol, albeit slower than WT (Fig. 6a). In contrast, we did not detect growth in butyric acid, oleic acid, or cholesterol as single carbon sources, corroborating our prediction (Fig. 6b–d).

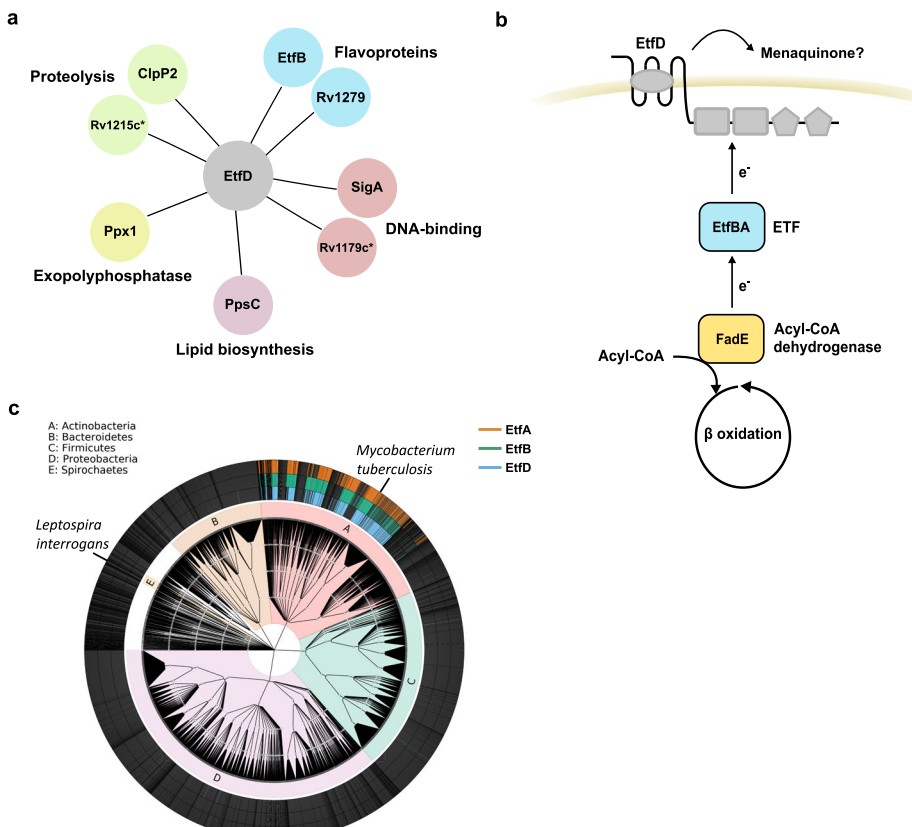

**Fig. 5 EtfD and EtfBA interaction and co-occurrence. a** EtfD cytoplasmic interactors identified by protein co-immunoprecipitation. **b** Model for the pathway constituted by EtfD and EtfBA. **c** Uniprot bacterial proteome database was surveyed for EtfD and EtfBA putative homologs. The inner ring corresponds to Phyla and the outer rings represent strains with a hit (>30% identity and >75% coverage) for EtfD, EtfB, or EtfA. Asterisk (*) in silico prediction.

To independently test the role of EtfBA$_{Mtb}$ and EtfD$_{Mtb}$ in β-oxidation, we assessed if an acyl-CoA oxidase (ACO) could allow both Δ*etfBA* and Δ*etfD* to grow with fatty acids. ACOs are peroxisomal enzymes that catalyze the same reaction as ACADs using molecular oxygen to re-oxidize FAD rather than ETF/ETFD[27]. The gene *pox3*, which encodes the extensively characterized ACO Pox3 from the yeast *Yarrowia lipolytica*[28,29] (Fig. 6d), was codon adapted for Mtb and expressed under the control of a strong, constitutive promoter in both Δ*etfBA* and Δ*etfD*. Pox3 has a high affinity for fatty acids with six and eight carbons chain length and low affinity for most other fatty acids[28]; thus, we grew strains in a medium with octanoic acid (high affinity), butyric acid (low affinity), or oleic acid (low affinity) as single carbon sources. Δ*etfD* was not able to utilize octanoic acid as single carbon source, further confirming its inability to oxidize multiple fatty acids, while the growth rate of Δ*etfD::pox3* was similar to that of wild type and complemented strain (Fig. 6e). A similar result was obtained with butyric acid, although it took longer for Δ*etfD::pox3* to reach wild type density (Supplementary Fig. 7). As expected, based on the affinity profile and function of Pox3, *pox3* did not rescue Δ*etfD* growth with oleic acid as single carbon source and it did not alter the growth rate on glycerol. Similarly, Δ*etfBA* did not grow on octanoic acid as single carbon source, while Δ*etfBA::pox3* was able to grow in the same medium, although reaching a lower final optical density than WT and complemented mutant. Δ*etfBA::pox3* was able to grow with butyric acid entering stationary phase 21 days after wild-type and complemented strain, it did not grow with oleic acid and it conferred a slight advantage on glycerol when compared with Δ*etfBA* (Fig. 6e and Supplementary Fig. 7). Acyl-CoA oxidase

activity was confirmed in the cell lysates of Pox3 expressing strains (Supplementary Fig. 8).

These results confirmed that EtfBA$_{Mtb}$ and EtfD$_{Mtb}$ constitute a complex necessary for the activity of fatty acid β-oxidation ACADs.

## Discussion

Mtb's energy metabolism displays a remarkable plasticity which supports its adaptation to a multitude of host microenvironments[30]. Its unusual domain structure suggested that the membrane protein encoded by *rv0338c* could be an unrecognized, essential component of Mtb's energy metabolism. Our studies identified Rv0338c as a member of a short electron transfer pathway essential for ACADs activity. Based on the similarity of this system to the human ETF system we propose to rename the respective Mtb genes as *etfD*$_{Mtb}$ (*rv0338c*), *etfB*$_{Mtb}$ (*rv3028c*), and *etfA*$_{Mtb}$ (*rv3029c*).

The increased susceptibility to fatty acids of an EtfD$_{Mtb}$ TetOff strain suggested a possible connection with fatty acid metabolism. Accordingly, our data showed that an *etfD*$_{Mtb}$ deletion mutant was not capable of utilizing fatty acids with four carbons or more as single carbon sources, thus indicating impairment in β-oxidation. We also found that Δ*etfD* cannot utilize cholesterol. These metabolic defects were associated with different outcomes regarding viability: butyric acid (short-chain) was not toxic, while palmitic acid and oleic acid (long-chain) were bactericidal with or without glycerol and cholesterol negatively impacted growth in mixed carbon source medium (glycerol + cholesterol). Importantly, Δ*etfD* did not grow and presented a survival defect in mice. Long-chain fatty acids, including oleic acid, are common

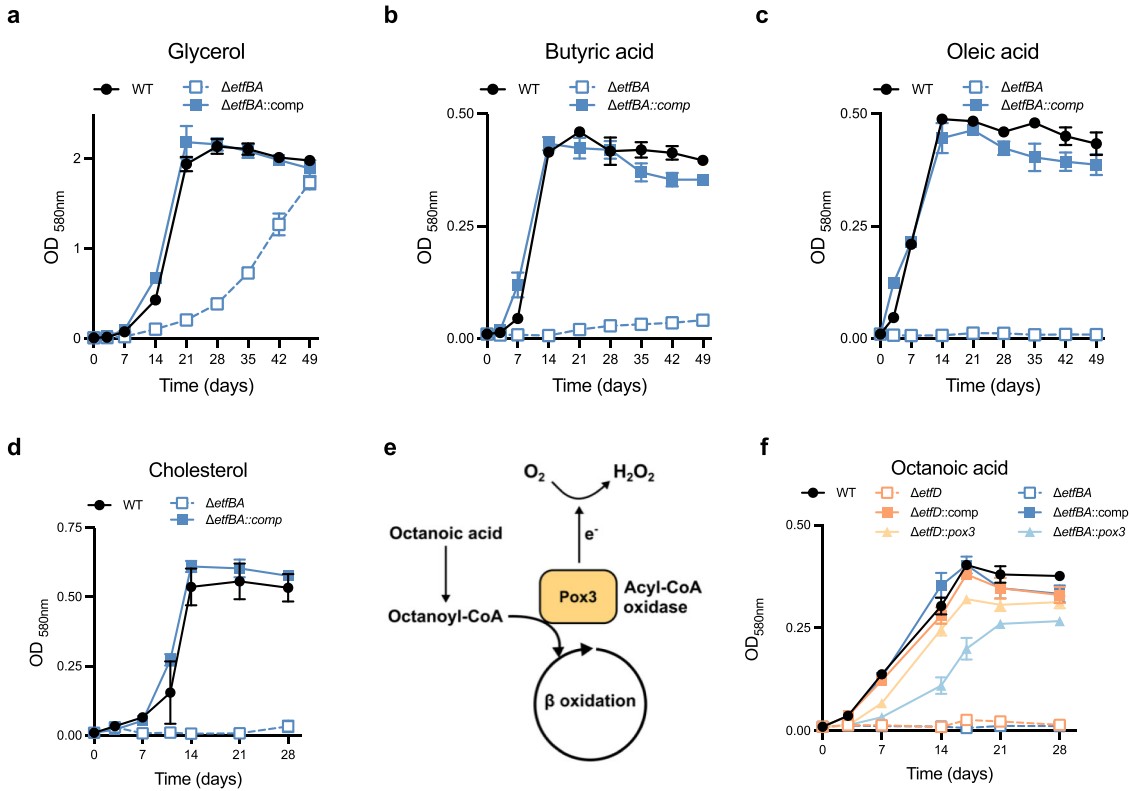

**Fig. 6 EtfD and EtfBA constitute a pathway necessary for the activity of acyl-coA dehydrogenases.** Growth with glycerol 25 mM (**a**), butyric acid 2.5 mM (**b**), oleic acid 250 μM (**c**), and cholesterol 250 μM (**d**) as single carbon sources. Oleic acid was replenished every 3–4 days for the first 14 days of culture to support growth and minimize toxicity. Cholesterol was replenished every to 3–4 days for the first 14 days to minimize precipitation. Data are averages of three replicates and are representative of three independent experiments. Error bars correspond to standard deviation. "Comp" stands for complemented. **e** Diagram representing acyl-coA oxidase activity on octanoic acid. **f** Rescue of Δ*etfD* and Δ*etfBA* growth with 250 μM octanoic acid as sole carbon source with the expression of the acyl-CoA oxidase Pox3. Octanoic acid was replenished every 3–4 days for the first 14 days of culture to support growth and minimize toxicity. Data are averages of three replicates and are representative of three independent experiments. Error bars correspond to standard deviation. "Comp" stands for complemented.

components of human macrophages[31] and carbon sources for Mtb. Similarly, cholesterol is a carbon source for Mtb during infection[4]. Hence, the inability to utilize fatty acids and cholesterol, together with the increased susceptibility to long-chain fatty acids likely explain the in vivo essentiality of EtfD$_{Mtb}$. The mechanism of long-chain fatty acid toxicity to Mtb is still poorly understood, even though it was first described in the 1940's[32]. In other bacteria, several mechanistic explanations have been proposed for the bactericidal activity of long-chain fatty acids, including membrane potential disruption[33,34], oxidative stress induction[35], or fatty acid biosynthesis inhibition[36]. Blocking Mtb cholesterol side chain β-oxidation was previously shown to impair cholesterol utilization and to result in the accumulation of toxic intermediates, which negatively impacted growth even in the presence of alternative carbon sources[7,12]. We have found that Δ*etfD* displays a similar phenotype, further reinforcing a function of EtfD in β-oxidation.

The metabolome of Δ*etfD* in medium with butyric acid as sole carbon source revealed an accumulation of butyryl-CoA, which indicated an impairment in β-oxidation. This can explain the inability of Δ*etfD* to oxidize fatty acids and connects EtfD$_{Mtb}$ with the β-oxidation enzymes that act on acyl-CoAs—the ACADs. None of Mtb's thirty-five annotated ACADs are essential in vivo[37] and there are no reports showing a specific ACAD being essential in vitro for the utilization of any fatty acid. Thus, ACADs are likely to be functionally redundant. However, this redundancy was not sufficient to support growth of Δ*etfD* in media with fatty acids as single carbon sources, strongly

suggesting that EtfD$_{Mtb}$ is necessary for the activity of multiple, if not all, ACADs.

Immunoprecipitation using EtfD$_{Mtb}$ as bait revealed multiple possible interacting proteins, suggesting that EtfD$_{Mtb}$ might integrate several pathways in Mtb. We were especially interested in the interaction with the cytoplasmic protein EtfB$_{Mtb}$, which together with EtfA$_{Mtb}$ constitute a putative electron transfer flavoprotein (ETF). Based on the data available for EtfB and EtfA homologs[38], these two proteins form a heterodimer. The fact that only EtfB was identified in the co-immunoprecipitation assay suggests that EtfD interacts more strongly with this subunit of Mtb's ETF. The human homologs (EtfAB) form an enzyme that re-oxidizes the FAD co-factor of multiple ACADs and transfers the electrons to the membrane bound oxidoreductase electron transfer flavoprotein oxidoreductase (EtfD), which then reduces the electron carrier ubiquinone, hence contributing to the generation of energy[39]. Mutations rendering defects in EtfAB or EtfD lead to a metabolic disease named multiple acyl-CoA dehydrogenase deficiency, which among other outcomes is characterized by the inability to oxidize fatty acids[40]. This led us to hypothesize that Mtb's EtfBA$_{Mtb}$ and EtfD$_{Mtb}$ might work together in a similar pathway. That EtfD$_{Mtb}$ and EtfBA$_{Mtb}$ show a strong pattern of co-occurrence across bacterial proteomes and Δ*etfBA* was unable to utilize fatty acids and cholesterol as single carbon sources were strong indications in favor of our hypothesis. Nevertheless, for a yet unclear mechanism, it was notable that Δ*etfBA* grew more slowly than Δ*etfD*. The expression of the acyl-CoA oxidase Pox3[28,29], an enzyme that catalyzes the same

reaction as ACADs, but uses molecular oxygen as an electron acceptor[27], was able to rescue the growth of both ΔetfD and ΔetfBA in fatty acids as single carbon source, showing that in both cases β-oxidation was impaired at the ACAD step and confirming our proposed hypothesis. Although we do not have direct evidence, it is very likely that the same phenomenon explains the inability of ΔetfD and ΔetfBA to utilize long chain fatty acids and cholesterol. Additionally, both mutants, with ΔetfBA displaying a stronger phenotype, presented a growth defect even with glycerol as single carbon source. It is possible that defective β-oxidation might interfere, for example, with mycolic acid recycling[41]. Also, ACADs participate in other metabolic pathways such as branched-chain amino acids catabolism[42], which might also have contributed to the growth defect in glycerol.

We are reporting a function for EtfD/EtfBA in β-oxidation, but it is possible that these proteins may have additional roles in Mtb. It has recently been reported that both EtfBA$_{Mtb}$ and EtfD$_{Mtb}$ are essential for resisting toxicity in media containing heme[43]. This further strengthened the functional relation between these proteins and suggested a role in iron metabolism. Interestingly, transcript levels of etfD$_{Mtb}$, but not etfBA$_{Mtb}$, respond to the iron content of the medium in an IdeR (iron metabolism transcriptional regulator) dependent manner[44,45]. It is also noteworthy, that the investigation of the Mtb transcriptional response to a series of compounds that inhibit EtfD[46] suggest a role in redox sensing. Curiously, it was shown that transcription of etfD is repressed by hydrogen peroxide[47]. It is thus possible that EtfD might also act as a redox sensor. Whether EtfBA$_{Mtb}$ and/or EtfD$_{Mtb}$ complex have a direct or indirect role in heme utilization and redox homeostasis is a question worthwhile to be addressed in the future.

In conclusion, we have identified a complex composed of EtfBA$_{Mtb}$ and the cognate membrane oxidoreductase EtfD$_{Mtb}$ that is required for the function of multiple ACADs. This complex constitutes a previously unsuspected vulnerable component of Mtb's β-oxidation machinery. EtfD$_{Mtb}$ has no structural homologs in humans and was recently proposed as the target of a series of compounds that are bactericidal against Mtb[46], which suggests that it could be a potential target for TB drug development. The presence of EtfD$_{Mtb}$ and EtfBA$_{Mtb}$ homologs in *Leptospira interrogans* suggests that this pathway might be relevant for other pathogens that rely on host fatty acids as carbon sources[26].

## Methods

**Culture conditions**. For cloning purposes we used *Escherichia coli* as a host, which was cultured in LB medium at 37 °C. Mtb was cultured at 37 °C in different media: Middlebrook 7H9 supplemented with 0.2% glycerol, 0.05% tyloxapol, and ADNaCl (0.5% fatty acid free BSA from Roche, 0.2% dextrose and 0.85% NaCl) or in Middlebrook 7H10 supplemented with 0.5% glycerol and 10% oleic acid-albumin-dextrose-catalase (OADC), and in a modified Sauton's minimal medium (0.05% potassium dihydrogen phosphate, 0.05% magnesium sulfate heptahydrate, 0.2% citric acid, 0.005% ferric ammonium citrate, and 0.0001% zinc sulfate) supplemented with 0.05% tyloxapol, 0.4% glucose, 0.2% glycerol, and ADNaCl with fatty-acid-free BSA (Roche). Modified Sauton's solid medium contained 1.5% bactoagar (BD) and glycerol at a higher concentration (0.5%). For single or mixed carbon source cultures we have used glycerol 25 mM, sodium acetate 2.5 mM, propionic acid 2.5 mM and butyric acid 2.5 mM. Octanoic acid, palmitic acid, oleic acid, and cholesterol were dissolved in a solution of tyloxapol and ethanol (1:1) and added to the medium at a final concentration of 250 μM to modified Sauton's minimal medium with 0.5% fatty acid free BSA (Roche) and 0.85% NaCl. Since fatty acids of eight carbons or more are toxic in the mM range, cultures were replenished every 3–4 days for the first 14 days, as described elsewhere[48]. We also used this strategy for cholesterol medium, due to its poor solubility. Tyloxapol and ethanol (1:1) was also added to the cultures with glycerol, acetic acid, propionic acid and butyric acid in the same quantities, to control for a possible impact on Mtb's growth. *Mycobacterium smegmatis* MC²155 was cultured in Middlebrook 7H10 supplemented with 0.2% glycerol at 37 °C. Antibiotics were used at the following final concentrations: carbenicillin 100 μg/ml, hygromycin 50 μg/ml and kanamycin 50 μg/ml.

**Mutant construction**. Mtb H37Rv etfD$_{Mtb}$ conditional knockdown was generated using a previously described strategy that control expression through proteolysis[49].

Briefly, a Flag tag and DAS + 4 tag were added to the 3′ end of etfD$_{Mtb}$, at the 3′ end of the target gene. This strain was then transformed with a plasmid expressing the adapter protein SspB under the control of TetR regulated promoters. In the presence of anhydrotetracycline (ATC) 500 μg/ml, TetR loses affinity to the promoter and sspB expression is de-repressed. SspB acts by delivering DAS + 4-tagged proteins to the native ClpXP protease. Hence, when ATC is added to the culture, SspB expression is induced and EtfD$_{Mtb}$-DAS-tag is degraded, working as a TetOFF system (EtfD$_{Mtb}$-TetOFF).

We have obtained deletion mutants through recombineering by using Mtb H37Rv expressing the recombinase RecET. Constructs with the hygromycin resistant gene (hygR) flanked by 500 bp upstream and downstream of the target loci were synthesized (GeneScript). In the case of etfD, since the flanking gene aspC is in the same orientation and it is essential for growth[50], we have included in the construct the constitutive promoter hsp60 to avoid polar effects. Mutants were selected in modified Sauton's with hygromycin. The plasmid expressing recET was counterselected by growing the deleted mutants in modified Sauton's supplemented with sucrose 10 %. For complementation we have cloned etfD and etfBA under the control of the promoter phsp60 into a plasmid with a kanamycin resistant cassette that integrates at the att-L5 site (pMCK-phsp60-etfD and pMCK-phsp60-etfBA) and transformed the deleted mutants. The gene pox3 from *Yarrowia lipolytica* was codon adapted for *Mtb* use, synthesized (GeneScript), cloned into a plasmid under the control of the promoter pTB38, with a kanamycin resistant cassette that integrates in the att-L5 site (pMCK-pTB38-pox3) and transformed into both etfD$_{Mtb}$ and etfBA$_{Mtb}$ deletion mutants. All generated strains and plasmids are listed in Supplementary Tables 3 and 4, respectively.

**Whole genome sequencing**. The genetic identity of ΔetfD and ΔetfBA was confirmed by whole genome sequencing (WGS).

Between 150 and 200 ng of genomic DNA was sheared acoustically and HiSeq sequencing libraries were prepared using the KAPA Hyper Prep Kit (Roche). PCR amplification of the libraries was carried out for ten cycles. $5–10 \times 10^6$ 50-bp paired-end reads were obtained for each sample on an Illumina HiSeq 2500 using the TruSeq SBS Kit v3 (Illumina). Post-run demultiplexing and adapter removal were performed and fastq files were inspected using fastqc[51]. Trimmed fastq files were then aligned to the reference genome (M. tuberculosis H37RvCO; NZ_CM001515.1) using bwa mem47. Bam files were sorted and merged using samtools48. Read groups were added and bam files de-duplicated using Picard tools and GATK best-practices were followed for SNP and indel detection49. Gene knockouts and cassette insertions were verified for all strains by direct comparison of reads spanning insertion points to plasmid maps and the genome sequence. Reads coverage data was obtained from the software Integrative Genomics Viewer version 2.5.2 (IGV)[52–54].

**PhoA fusion assay**. Truncated versions of EtfD$_{Mtb}$ were fused with the *E. coli* alkalyne phosphatase PhoA. This enzyme requires the oxidative environment of the periplasm to be active and degrades the substrate BCIP generating a blue precipitate. We have fused phoA at different residues located in the transmembrane domains. Positive control consisted in PhoA fused with the antigen 85B, while the negative control was PhoA alone[55]. All plasmids were transformed in *M. smegmatis* MC²155 and the assay was performed in LB plates with and without BCIP.

**Mouse infection**. Mouse experiments were performed in accordance with the Guide for the Care and Use of Laboratory Animals of the National Institutes of Health, with approval from the Institutional Animal Care and Use Committee of Weill Cornell Medicine. Forty-eight female, eight-week-old *Mus musculus* C57BL/6 (Jackson Labs) were infected with ~100–200 CFU/mouse using an Inhalation Exposure System (Glas-Col). Strains were grown to mid-exponential phase and single-cell suspensions were prepared in PBS with 0.05% Tween 80, and then resuspended in PBS. Lungs and spleen were homogenized in PBS and plated on modified Sauton's medium to determine CFU/organ at the indicated time points.

**Metabolomics**. Strains were grown in modified Sauton's until an OD$_{580nm}$ of 1 and 1 ml of culture was used to seed filters[56] placed on top of solid modified Sauton's medium. Bacteria grew for 7 days, after which the filters were transferred to solid modified Sauton's medium with butyric acid or $^{13}$C-labelled butyric acid (Cambridge Isotope Laboratories, Inc) at a final concentration of 2.5 mM for 24 h. For metabolite extraction bacteria were disrupted by bead beating three cycles, 50 s (Precellys 24, Bertin technologies) in a solution of acetonitrile:methanol:water (4:4:2). The relative abundances of butyric acid, CoA species and TCA intermediates were determined using an ion-pairing LC-MS system, as previously described[57]. In brief, samples (5 uL) were injected onto a ZORBAX RRHD Extend-C18 column (2.1 × 150 mm, 1.8 μm; Agilent Technologies) with a ZORBAX SB-C8 (2.1 mm × 30 mm, 3.5 μm; Agilent Technologies) precolumn heated to 40 °C and separated using a gradient of methanol in 5 mM tributylamine/5.5 mM acetate. Post-column, 10% dimethyl sulfoxide in acetone (0.2 ml/min) was mixed with the mobile phases to increase sensitivity. Detection was performed from m/z 50-1100, using an Agilent Accurate Mass 6230 Time of Flight (TOF) spectrometer with Agilent Jet Stream electrospray ionization source operating in the negative

ionization mode. Incorporation of $^{13}C$ was quantitated and corrected for natural $^{13}C$ abundance using Profinder B.08.00 (Agilent Technologies).

**Immunoprecipitation.** We transformed $\Delta etfD$ with a plasmid expressing flag tagged EtfD under hsp60 promoter (pMEK-Phsp60-etfD$_{Mtb}$-flag) and used WT Mtb expressing only the flag tag as a control. Mtb whole-cell lysates were collected from 120 ml log phase culture in butyric acid single carbon source Sauton's medium, incubated with 1% DDM for 2 h on ice, followed by anti-Flag beads (Sigma) overnight incubation with gentle rotation. Beads were collected on the second day, washed with lysis buffer (50 mM Tris-HCl, 50 mM NaCl, pH 7.4), and eluted with 100 ng/µl Flag peptide. The eluates were resolved on SDS-PAGE before mass spectrometry.

For mass spectrometry analysis, the total spectrum count (TSC) from biological duplicates were summed. We calculated the ratio of summed TSC from EtfD$_{Mtb}$-Flag vs. Flag control and used a cut-off of $\geq 10$.

**Acyl-CoA oxidase activity in cell lysates.** Mtb strains were grown in modified Sauton's minimal medium until mid-exponential phase (OD ~0.5). Bacteria were lysed by bead beating three cycles, 50 s (Precellys 24, Bertia technologies) in 11 mM potassium phosphate buffer (pH 7.4) with protease inhibitor (cOmplete™, Mini Protease Inhibitor Cocktail; Roche). Samples were concentrated with Pierce™ Protein Concentrators PES, 3K MWCO (ThermoFisher Scientific) and resuspend in the same buffer to exclude metabolites that could interfere with the reaction. The acyl-CoA oxidase assay was adapted from previous reports[58,59]. Briefly, the assay mixture contained 11 mM potassium phosphate buffer (pH 7.4), 40 mM amino-triazole (Sigma-Aldrich), 0.04 mg/ml peroxidase from horseradish Type VI-A (Sigma-Aldrich), 104 µM 2′,7′-dichlorofluorescin diacetate (Sigma-Aldrich) and 30 µM octanoyl coenzyme A lithium salt hydrate (Sigma-Aldrich). The assays were performed in black 96 well-plates with clear bottom (Costar) with a sample volume of 100 µl and a total volume of 200 µl. Fluorescence (excitation, 503 nm; emission, 529 nm) was recorded for 10 min, and the activity was expressed as 2′,7′-dichlor-ofluorescein (DCF) nmol produced/min mg of protein. For quantification, we generated a standard curve for DCF (Sigma-Aldrich) dissolved 11 mM potassium phosphate buffer (pH 7.4). Protein was quantified using the Qubit™ Protein Assay Kit (ThermoFisher Scientific).

**In silico analysis.** Transmembrane domain topology of EtfD$_{Mtb}$ was performed in MEMSAT3[21]. Domain architecture of EtfD$_{Mtb}$, EtfB and EtfA was based on HHPred[60] and XtalPred-RF[61]. Eggnog[62] was used for the cluster of orthologous groups analysis (COG). The members of COG247 that include *etfD* were used to generate a rootless phylogenetic tree in iTOL (version 6)[63].

To analyze the presence or absence of EtfD$_{Mtb}$, EtfB$_{Mtb}$, and EtfA$_{Mtb}$ homologs across bacterial species, we obtained the set of UniProt reference bacterial proteomes, which are selected both manually and algorithmically by UniProt as landmarks in (bacterial) proteome space[64]. We discarded proteomes with no taxonomic labels and performed the analysis on a final set of 6240 bacterial reference proteomes. Using EtfD$_{Mtb}$, EtfB$_{Mtb}$, and EtfA$_{Mtb}$ as query protein sequences, we used the following protein BLAST (BLASTp) parameter values: identity cutoff of >30%, coverage cutoff of >75%, $e$-value cutoff of 10-3. Visual representations of phylogenies with surrounding color-coded rings were generated using the software tool GraPhlAn[65], with the phylogenetic try built from the taxonomic categorization of the 6240 UniProt bacterial reference proteomes. To evaluate the statistical significance of co-occurrence of EtfD$_{Mtb}$ and EtfBA$_{Mtb}$ in Actinobacteria, we performed a hypergeometric test to evaluate the probability of observing k species with EtfD and EtfBA homologs, given M Actinobacterial species, n species with EtfD homologs, and N species with EtfBA homologs.

**Quantification and statistical analysis.** Generation of graphics and data analyses were performed in Prism version 9.0 software (GraphPad).

**Reporting summary.** Further information on research design is available in the Nature Research Reporting Summary linked to this article.

## Data availability
Whole genome sequencing data generated in this study were deposited in NCBI's Sequence Read Archive (SRA) under BioProject PRJNA670664. LC-MS data generated in this study were deposited in the MetaboLights database[66] under accession code MTBLS2374. To identify possible homologues of EtfD, EtfB, and EtfA, we used the subset of Uniprot reference proteomes corresponding to bacteria (https://www.uniprot.org/proteomes/?query=*&fil=taxonomy%3A%22Bacteria+%5B2%5D%22+AND+reference%3Ayes). Source data for figures are provided as a Source Data file. Source data are provided with this paper.

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

## Acknowledgements

We thank J. McConnell and C. Trujillo for help with strain generation and in vivo characterization. We thank K.G. Papavinasasundaram (University of Massachusetts), R. Aslebagh and S.A. Shaffer (University of Massachusetts, Mass Spectrometry Facility) for LC-MS/MS analysis. We acknowledge J.M. Bean from MSKCC and the use of the Integrated Genomics Operation Core at MSKCC, funded by the NCI Cancer Center Support Grant (CCSG, P30 CA08748), Cycle for Survival and the Marie-Josée and Henry R. Kravis Center for Molecular Oncology. D.S. and S.E. were supported by Tri-Institutional TB Research Unit U19AI111143. T.B. was supported by a Potts Memorial Foundation fellowship.

## Author contributions

S.E. and D.S. conceived ideas, supervised the study and revised the manuscript. T.B. conceived ideas, performed experimental work, analysed and interpreted data and wrote the manuscript. R.S.J. and R.W. performed experimental work, analysed and interpreted data and reviewed the manuscript. A.J. performed the in silico analysis and reviewed the manuscript. K.Y.R. supervised the metabolomics experiments and reviewed the manuscript.

## Competing interests

The authors declare no competing interests.
