## [Peer Review File · Nature Communications]

Multiple acyl-CoA dehydrogenase deficiency kills
Mycobacterium tuberculosis in vitro and during infectionREVIEWER COMMENTS

Reviewer #1 (Remarks to the Author):

The manuscript describes the findings by the authors which reveal the role of a previously uncharacterized protein complex in *Mycobacterium tuberculosis* (Mtb) in the β -oxidation of fatty acids. Since Mtb is thought to utilize fatty acids inside the human body, the findings are highly significant. Furthermore, the EtfD protein in Mtb could be a target for drugs since it does not have structural homologs in the human body.

The protein encoded by Rv0338 (EtfMtb) is a membrane protein which has not been studied before and *in silico* analysis in this study reveals it to be a chimeric protein containing domains that are similar to those found in dehydrogenases and reductases. The authors analyzed the protein's function by conditional knockdown of the protein via proteolysis and by deletion of the *etfD* gene. They show that in the absence of the EtfD protein, Mtb was unable to grow on medium containing oleic acid and in mouse lungs. They show that long-chain fatty acids were bactericidal for the mutant strain. Using metabolomics, the authors show that loss of EtfD was detrimental to the function of acyl-CoA dehydrogenases resulting in impaired catabolism of fatty acids. They show that EtfD interacts with EtfB by immunoprecipitation. They also generated an Mtb strain lacking EtfBA and show that it was not able to grow on octanoic acid as sole carbon source. The authors show strong evidence for their claim on the requirement of both EtfBA and EtfD for ACAD activity *in vitro*. They claim that the EtfBA-EtfD complex is necessary for β -oxidation of fatty acids by acyl-CoA dehydrogenases.

The claims and conclusions are mostly supported by the data shown. However, the claim on line 152 that EtfBA is also required for ACAD activity along with EtfD is not verified *in vivo* as done for EtfD (Fig. 2). The authors show data on their analysis of the Δ *etfBA* mutant *in vitro* but not *in vivo*. Since they show *in vitro* analyses of the Δ *etfBA* mutant (Figs. 6, S7), why have the authors not performed a test of *in vivo* essentiality of EtfBA like they have done for the Δ *etfD* mutant? Is EtfBA essential for fatty acid β -oxidation *in vivo*?

The findings reported in this study on the roles of EtfD and EtfBA add to the previously reported findings by other groups on the potential roles of the proteins in iron metabolism. The authors acknowledge that these proteins could have other functions additional to those analyzed in this study. Since this protein complex has not been studied before, the publication of this study is likely to stimulate further studies on this protein complex and its role in the *in vivo* nutrition and persistence of Mtb.

The experimental methodology is sound, the work is of a high standard and follows a previously published study by the same group in this journal. Details provided in the methods would be sufficient for the work to be reproduced.

Reviewer #2 (Remarks to the Author):

In this study, Beites et al. propose that Rv0338c, a protein expressed by *Mycobacterium tuberculosis* (Mtb) that is currently annotated as a probable iron-sulfur-binding reductase, functions as a membrane dehydrogenase required for fatty acid β -oxidation. While the essentiality of *rv0338c* for Mtb growth on fatty acids *in vitro* and survival *in vivo* are novel and important findings, there is a lack of direct evidence that Rv0338c operates as a dehydrogenase and that EtfAMtb/EtfBMtb constitute Mtb's electron transfer flavoprotein (ETF), and a number of issues to clarify.

Sup. Figure 1: The proposed topology for Rv0338c (Fig. S1a), based on experiments using PhoA fusion constructs, differs from that predicted in a recent study (Székely et al, 2020) by the absence of a sixth transmembrane domain. Importantly, the C-terminal portion of the protein is located in the cytoplasm in the present study, while it is predicted to face the periplasm in (Székely et al, 2020). According to the MEMSAT topology prediction shown in Sup Fig. 1b, PhoA

fusions at residues L198 and Q203 should face the periplasm and therefore react with BCIP, but they do not in Sup Fig. 1a. Moreover, only 8/49 of the proteins immunoprecipitated with Rv0338c are predicted to be located in the cytoplasm. Consequently, additional PhoA fusions at residues in the 258-882 region, or alternative approaches, should be used to demonstrate convincingly that the C-terminal portion of Rv0338c faces the cytoplasm, as stated.

No information with regard to how bacterial cultures were supplemented with fatty acids is provided. It is important to know whether fatty acids were complexed with albumin prior to addition to the cultures, and how palmitic and oleic acids were replenished (as free fatty acids, as fatty acids pre-complexed to albumin, or by renewing fatty acid-supplemented culture medium) in order to compare the various culture conditions in Figure 3 a and b. Possible links between long chain fatty acid-driven toxicity and the dehydrogenase function of Rv0388c should be discussed.

The hypothesis that Mtb acyl-CoA dehydrogenase activity requires Rv0388c relies essentially on the observation of a block in β -oxidation of butyric acid in the rv0338c Mtb mutant grown on labelled butyric acid as a single carbon source (Figure 4), and inability of this mutant to grow on fatty acids with four carbons and more (Figure 3). While very clear and reproducible, the shown accumulation of butyryl-CoA is insufficient to conclude on a loss of ACAD activity in the rv0338c Mtb mutant. A dedicated assay of ACAD activity would be required. Moreover, to extend this finding to fatty acids with more than four carbons (L. 190), assays of ACAD activity should be performed with at least one longer chain fatty acid.

Based on their Rv0338c (EtfDMtb) interactome analysis, the authors propose that EtfDMtb interacts with EtfAMtb and EtfBMtb, a hypothesis supported by the co-occurrence of EtfAMtb, EtfBMtb and EtfDMtb across bacterial proteomes, and the immunoprecipitation of EtfBMtb with EtfDMtb (Sup. Table 2). Why EtfAMtb was not immunoprecipitated with EtfDMtb should be discussed. Experiments showing that EtfBAMtb mutants are, like EtfDMtb mutants, unable to grow on fatty acids, and that the growth defects of both mutants are rescued by expression of Pox3 provide a strong argument for EtfAMtb and EtfBMtb constituting a complex required for fatty acid β -oxidation. However, proofs that Pox3 functions as an acyl-CoA oxidase in the context of Mtb, and that EtfAMtb and EtfBMtb are Mtb's ETF are lacking. Székely et al. recently reported a gene expression analysis of Mtb exposed to a chemical inhibitor of Rv0338c, which suggested that this protein acts as a redox sensor (Székely et al, 2020). The authors should discuss this hypothesis, and attempt to reconcile these data with their own findings. A comparison of the transcriptomes of wild-type, EtfDMtb mutant and complemented strain, may be conducted to consolidate the hypothesis that Rv0338c functions as an ETF dehydrogenase.

Minor comments

L35: Qualify the statement that inactivation of Mce1 reduces intracellular growth, considering discordant results reported by (Shimono et al, 2003).

Figure 1c and Sup Figure 2: Quantification of the 3 spot assays should be shown next to representative pictures. While growing better than in 7H10 + OADC, the growth of the knock-out strain is inferior to that of the WT strain in 7H10 + fatty acid free ADN (Sup Figure 2). The statement L87 should therefore be corrected accordingly. Similarly in Figure 3a, the mutant grows less well in glycerol than the WT strain. The defective growth of the mutant in the presence of such carbon sources should be discussed, in light of the proposed function of Rv0338c.

L154: Correct "impair the of use"

L285-290: Errors in references

References

Shimono N, Morici L, Casali N, Cantrell S, Sidders B, Ehrt S & Riley LW (2003) Hypervirulent mutant of Mycobacterium tuberculosis resulting from disruption of the mce1 operon. Proc. Natl.

Acad. Sci. 100: 15918–15923 Available at:
<http://www.pnas.org/cgi/doi/10.1073/pnas.2433882100>

Székely R, Rengifo-Gonzalez M, Singh V, Riabova O, Benjak A, Piton J, Cimino M, Kornobis E, Mizrahi V, Johnsson K, Manina G, Makarov V & Cole ST (2020) 6,11-Dioxobenzo[f]pyrido[1,2-a]indoles Kill Mycobacterium tuberculosis by Targeting Iron-Sulfur Protein Rv0338c (IspQ), A Putative Redox Sensor. ACS Infect. Dis.

Reviewer #3 (Remarks to the Author):

The manuscript by Beites et al. describes the role of oxidoreductases that function in the β -oxidation of fatty acids in Mycobacterium tuberculosis (Mtb). More specifically, the authors use molecular genetics approaches to identify EtfD and EtfAB as electron transfer components that function with acyl-CoA dehydrogenases (ACADs) and further demonstrated the importance of EtfD in pathogenesis using an infection model. The major conclusions are novel and should be of broad interest. For the most part, the manuscript is well-written, the data are clearly presented, and the data are convincing. Some additional experiments would strengthen the conclusions and broaden the significance of the work.

Specific comments:

1. The authors should refer to EtfD as an oxidoreductase, not a dehydrogenase (Abstract and elsewhere). The formal definition of a dehydrogenase is an enzyme that catalyzes the removal of hydrogen atoms from a substrate. The authors have not demonstrated this for EtfD. The dehydrogenase in the system is the ACAD. EtfD and EtfAB appear to be electron transfer components that enable the ACAD to turnover. This is clear in Figure 5, but is especially confusing in the Abstract.
2. In stating that cholesterol degradation yields acetyl-CoA, propionyl-CoA, succinyl-CoA and pyruvate, the authors cite a review article (II. 40-41). They should cite the original literature.
3. Considering that cholesterol catabolism is featured in the Introduction and that this catabolism depends on ACADs, the authors should test whether the etfD and etfAB mutants grow on cholesterol. More specifically, ACADs are involved in degrading both the cholesterol side chain and some of the rings (Thomas and Sampson. 2013. Biochemistry 52, 2895-2904. DOI: 10.1021/bi4002979; Gadbery et al. 2020. Biochemistry 59, 1113-1123. doi: 10.1021/acs.biochem.0c00005). Evaluating the involvement of the Etf proteins in cholesterol catabolism would broaden the significance of this study. In addition, establishing whether growth on cholesterol is impaired would provide a better context for interpreting the data obtained in the infection model.
4. The conclusions that EtfAB forms a complex and is a flavoprotein should be strengthened (II. 211-212). Related to this, it is noted that only EtfB co-precipitated with EtfD. Specifically, the authors should produce EtfAB heterologously. The proteins appear to be cytoplasmic, so if one component was produced with a His-tag, it should be quite straightforward to demonstrate that the two proteins form a single, multimeric complex that contains flavin.
5. In Figure 5b, the authors should indicate that the system requires a terminal electron acceptor (i.e., the equivalent of ubiquinone that serves this function for the human homologs). Depicting a generic requirement would suffice.

We thank the editor and the reviewers for their positive feedback and helpful comments. We have addressed the specific points raised in their reviews. Below, please find the reviewers' comments, followed by our responses.

Reviewer #1 (Remarks to the Author):

The manuscript describes the findings by the authors which reveal the role of a previously uncharacterized protein complex in *Mycobacterium tuberculosis* (Mtb) in the β -oxidation of fatty acids. Since Mtb is thought to utilize fatty acids inside the human body, the findings are highly significant. Furthermore, the EtfD protein in Mtb could be a target for drugs since it does not have structural homologs in the human body.

The protein encoded by Rv0338 (EtfMtb) is a membrane protein which has not been studied before and in silico analysis in this study reveals it to be a chimeric protein containing domains that are similar to those found in dehydrogenases and reductases. The authors analyzed the protein's function by conditional knockdown of the protein via proteolysis and by deletion of the *etfD* gene. They show that in the absence of the EtfD protein, Mtb was unable to grow on medium containing oleic acid and in mouse lungs. They show that long-chain fatty acids were bactericidal for the mutant strain. Using metabolomics, the authors show that loss of EtfD was detrimental to the function of acyl-CoA dehydrogenases resulting in impaired catabolism of fatty acids. They show that EtfD interacts with EtfB by immunoprecipitation. They also generated an Mtb strain lacking EtfBA and show that it was not able to grow on octanoic acid as sole carbon source. The authors show strong evidence for their claim on the requirement of both EtfBA and EtfD for ACAD activity in vitro. They claim that the EtfBA-EtfD complex is necessary for β -oxidation of fatty acids by acyl-CoA dehydrogenases.

The claims and conclusions are mostly supported by the data shown. However, the claim on line 152 that EtfBA is also required for ACAD activity along with EtfD is not verified in vivo as done for EtfD (Fig. 2). The authors show data on their analysis of the Δ *etfBA* mutant in vitro but not in vivo. Since they show in vitro analyses of the Δ *etfBA* mutant (Figs. 6, S7), why have the authors not performed a test of in vivo essentiality of EtfBA like they have done for the Δ *etfD* mutant? Is EtfBA essential for fatty acid β -oxidation in vivo?

The question raised by the reviewer is very important. In the present work we focused on the functionalization of EtfD and validated it as a potential drug target. We are now working on a comprehensive functional analysis of Δ *etfBA*. We are currently determining the phenotype of Δ *etfBA* in mice, and so far, we have only recovered colon forming units (CFUs) from lungs of mice infected with wild-type and complemented strain. We determined the viability of the bacilli in the single cell suspension used to infect the mice, and Δ *etfBA* cells were viable before mouse infection. Thus, we hypothesize that lung homogenate-derived lipids are inhibiting the growth of Δ *etfBA* on plates. We will try to recover CFU by making higher dilutions of the lung homogenates. At this point, unfortunately, we will not be able to finish 2 independent experiments in an adequate time frame for this resubmission. Nevertheless, we can share the histopathology analysis of lungs at day 28 (end of acute phase of infection) where lesions in the lungs of mice infected with wild-type and complemented strains can be

observed while such lesion are absent in lungs from mice infected with $\Delta etfBA$ (Fig. 1). These preliminary data suggest that EtfBA are also required for growth in vivo.

Fig. 1 Mouse lung sections stained with hematoxylin and eosin. A) Scans of lung sections stained with H&E from mice infected with wild-type, $\Delta etfBA$ and $\Delta etfBA::$ complemented mutant. B) Micrographs of lung lesions from mice infected with wild-type, mutant and $\Delta etfBA::$ complemented mutant (2x amplification).

The findings reported in this study on the roles of EtfD and EtfBA add to the previously reported findings by other groups on the potential roles of the proteins in iron metabolism. The authors acknowledge that these proteins could have other functions additional to those analyzed in this study. Since this protein complex has not been studied before, the publication of this study is likely to stimulate further studies on this protein complex and its role in the in vivo nutrition and persistence of Mtb.

The experimental methodology is sound, the work is of a high standard and follows a previously published study by the same group in this journal. Details provided in the methods would be sufficient for the work to be reproduced.

Thank you for the positive comments.

Reviewer #2 (Remarks to the Author):

In this study, Beites et al. propose that Rv0338c, a protein expressed by *Mycobacterium tuberculosis* (Mtb) that is currently annotated as a probable iron-sulfur-binding reductase, functions as a membrane dehydrogenase required for fatty acid β -oxidation. While the essentiality of rv0338c for Mtb growth on fatty acids in vitro and survival in vivo are novel and important findings, there is a lack of direct evidence that Rv0338c operates as a dehydrogenase and that EtfAMtb/EtfBMtb constitute Mtb's electron transfer flavoprotein (ETF), and a number of issues to clarify.

Sup. Figure 1: The proposed topology for Rv0338c (Fig. S1a), based on experiments using PhoA fusion constructs, differs from that predicted in a recent study (Székely et al, 2020) by the absence of a sixth transmembrane domain. Importantly, the C-terminal portion of the protein is located in the cytoplasm in the present study, while it is predicted to face the periplasm in (Székely et al, 2020). According to the MEMSAT topology prediction shown in Sup Fig. 1b, PhoA fusions at residues L198 and Q203 should face the periplasm and therefore react with BCIP, but they do not in Sup Fig. 1a. Moreover, only 8/49 of the proteins immunoprecipitated with Rv0338c are predicted to be located in the cytoplasm. Consequently, additional PhoA fusions at residues in the 258-882 region, or alternative approaches, should be used to demonstrate convincingly that the C-terminal portion of Rv0338c faces the cytoplasm, as stated.

We agree with the reviewer regarding the importance of determining where the C-terminal portion of Rv0338c is located. To further confirm our results, we have generated 2 additional fusions with PhoA: E400 and L597. *M. smegmatis* expressing both fusions did not catabolize BCIP, which further showed that the C-terminal portion of Rv0338c is facing the cytoplasm. These new data have been added to the revised Supplementary Fig. 1.

No information with regard to how bacterial cultures were supplemented with fatty acids is provided. It is important to know whether fatty acids were complexed with albumin prior to addition to the cultures, and how palmitic and oleic acids were replenished (as free fatty acids, as fatty acids pre-complexed to albumin, or by renewing fatty acid-supplemented culture medium) in order to compare the various culture conditions in Figure 3 a and b.

Fatty acids of 8 carbons or more were dissolved in a solution of tyloxapol and ethanol in a proportion of 1:1. The tyloxapol:ethanol stocks of fatty acids were added to the media which contains albumin in the beginning of the culture and for replenishment. We have also added an equal amount of tyloxapol:ethanol to the media with glycerol, acetic acid, propionic acid and butyric acid as single carbon sources to control for a possible impact on growth. We have altered the methods section accordingly (please see lines 296-303). We thank the reviewer for raising this point.

Possible links between long chain fatty acid-driven toxicity and the dehydrogenase function of Rv0388c should be discussed.

The inability to oxidize long chain fatty acids might have pleiotropic effects, which makes it hard to pinpoint a specific mechanism for toxicity in Mtb. Nevertheless, in the discussion section we have referred to some possible long chain fatty acid toxicity mechanisms in other bacteria. In addition, blocking Mtb cholesterol side chain β -oxidation was previously shown to result in the accumulation of toxic intermediates (please see lines 222-229). At this point we unfortunately do not have additional data that allow us to further speculate about the possible link between fatty acid toxicity and oxidoreductase activity of Rv0388/EtfD.

The hypothesis that Mtb acyl-CoA dehydrogenase activity requires Rv0388c relies essentially on the observation of a block in β -oxidation of butyric acid in the rv0338c Mtb mutant grown on labelled butyric acid as a single carbon source (Figure 4), and inability of this mutant to grow on fatty acids with four carbons and more (Figure 3). While very clear and reproducible, the shown accumulation of butyryl-CoA is insufficient to conclude on a loss of ACAD activity in the rv0338c Mtb mutant. A dedicated assay of ACAD activity would be required.

We agree with the reviewer, and during this work, we had considered ACAD activity assay, but we believe that it would not reveal if EtfD/EtfBA are required for ACAD activity. This is because the available ACAD enzymatic assays rely on adding an electron acceptor, for example ferrocenium hexafluorophosphate (Wipperman MF, 2013 PMID: 23836861), to re-oxidize the FAD co-factor of ACADs. Since we propose that EtfD/EtfBA function is to re-oxidize the FAD co-factor of ACADs, the presence of ferrocenium hexafluorophosphate in the assay would likely bypass the need of EtfD/EtfBA, and we would likely not detect ACAD enzymatic activity impairment in the mutant. Thus, with the current available methods and knowledge, we hope that the reviewer agrees with our view that the metabolomics data are the best approach to show an impact on ACADs.

Moreover, to extend this finding to fatty acids with more than four carbons (L. 190), assays of ACAD activity should be performed with at least one longer chain fatty acid.

We believe that the rescue of Δ etfB and Δ etfBA growth in media with butyric acid and octanoic acid as single carbon sources with the expression of Pox3 demonstrates (i) that the activity of ACADs operating in the degradation of these two fatty acids are restricting growth, and (ii) that this phenotype is not restricted to butyric acid. Nevertheless, we agree that there are no data regarding ACAD activity in media with longer fatty acids (oleic acid and palmitic acid) and modified the discussion section accordingly (please see lines 258-264).

Based on their Rv0338c (EtfDMtb) interactome analysis, the authors propose that EtfDMtb interacts with EtfAMtb and EtfBMtb, a hypothesis supported by the co-occurrence of EtfAMtb, EtfBMtb and EtfDMtb across bacterial proteomes, and the immunoprecipitation of EtfBMtb with EtfDMtb (Sup. Table 2). Why EtfAMtb was not immunoprecipitated with EtfDMtb should be discussed.

It is possible that EtfD interacts more strongly with the EtfB subunit of Mtb's ETF complex. We added this hypothesis to the discussion section (please see lines 225-226). However, without structural data this remains a hypothesis, especially, because EtfDMtb is not a homologue of the human ETFD, or of other described ETF oxidoreductases, for which structural studies are available. We intend, as a follow up work, to get into more detail on physical interactions of EtfD-EtfBA, the heterodimer EtfBA itself, and also EtfBA-ACADs. We believe that these studies are beyond the scope of this manuscript, and we hope the reviewer agrees with our assessment.

Experiments showing that EtfBAMtb mutants are, like EtfDMtb mutants, unable to grow on fatty acids, and that the growth defects of both mutants are rescued by expression of Pox3 provide a strong argument for EtfAMtb and EtfBMtb constituting a complex required for fatty acid β -oxidation. However, proofs that Pox3 functions as an acyl-CoA oxidase in the context of Mtb, and that EtfAMtb and EtfBMtb are Mtb's ETF are lacking.

We thank the reviewer for raising this point. We have adapted a protocol to measure acyl-coA oxidase activity in crude extracts and we were able to show that the strains expressing Pox3 show acyl-coA oxidase activity when octanoyl-CoA is present in the assay. Please see the newly added Supplementary Fig.8, results section (lines 198-199) and methods section (lines 392-408)

Székely et al. recently reported a gene expression analysis of Mtb exposed to a chemical inhibitor of Rv0338c, which suggested that this protein acts as a redox sensor (Székely et al, 2020). The authors should discuss this hypothesis, and attempt to reconcile these data with their own findings. A

comparison of the transcriptomes of wild-type, EtfDMtb mutant and complemented strain, may be conducted to consolidate the hypothesis that Rv0338c functions as an ETF dehydrogenase.

Consistent with the findings of Széleky et al. transcription of *etfD* is repressed by hydrogen peroxide (Schnappinger D, 2003 PMID: 1295309), which supports a possible role of EtfD in Mtb redox homeostasis. We added this hypothesis to the discussion section (see lines 270-275).

In the follow up experiments for this line of research, we are assessing if this complex acts on other physiological processes, and for that we will perform RNA seq. We hope that the reviewer agrees that the RNA seq experiment goes beyond the scope of the present work.

Minor comments

L35: Qualify the statement that inactivation of Mce1 reduces intracellular growth, considering discordant results reported by (Shimono et al, 2003).

Thank you for raising this issue. The introduction section was changed accordingly (please see line 38-39).

Figure 1c and Sup Figure 2: Quantification of the 3 spot assays should be shown next to representative pictures.

Quantifications of the spot assays have been added.

While growing better than in 7H10 + OADC, the growth of the knock-out strain is inferior to that of the WT strain in 7H10 + fatty acid free ADN (Sup Figure 2). The statement L87 should therefore be corrected accordingly. Similarly in Figure 3a, the mutant grows less well in glycerol than the WT strain. The defective growth of the mutant in the presence of such carbon sources should be discussed, in light of the proposed function of Rv0338c.

We thank the reviewer for requesting further discussion of this phenotype. Although we do not have direct evidence, it is possible that ACADs participate in other metabolic pathways, like cell wall recycling or branched chain amino acid catabolism which may be compromised, contributing to the growth defects in medium without fatty acids. We have added this hypothesis to the discussion section (please see lines 260-264)

L154: Correct "impair the of use"

The manuscript has been revised accordingly.

L285-290: Errors in references

References were corrected.

References

Shimono N, Morici L, Casali N, Cantrell S, Sidders B, Ehrt S & Riley LW (2003) Hypervirulent mutant of *Mycobacterium tuberculosis* resulting from disruption of the *mce1* operon. *Proc. Natl. Acad. Sci.* 100: 15918–15923 Available at: <http://www.pnas.org/cgi/doi/10.1073/pnas.2433882100>

Székely R, Rengifo-Gonzalez M, Singh V, Riabova O, Benjak A, Piton J, Cimino M, Kornobis E, Mizrahi V, Johnsson K, Manina G, Makarov V & Cole ST (2020) 6,11-Dioxobenzo[f]pyrido[1,2- a]indoles Kill Mycobacterium tuberculosis by Targeting Iron-Sulfur Protein Rv0338c (IspQ), A Putative Redox Sensor. ACS Infect. Dis.

Reviewer #3 (Remarks to the Author):

The manuscript by Beites et al. describes the role of oxidoreductases that function in the β -oxidation of fatty acids in Mycobacterium tuberculosis (Mtb). More specifically, the authors use molecular genetics approaches to identify EtfD and EtfAB as electron transfer components that function with acyl-CoA dehydrogenases (ACADs) and further demonstrated the importance of EtfD in pathogenesis using an infection model. The major conclusions are novel and should be of broad interest. For the most part, the manuscript is well-written, the data are clearly presented, and the data are convincing. Some additional experiments would strengthen the conclusions and broaden the significance of the work.

Specific comments:

1. The authors should refer to EtfD as an oxidoreductase, not a dehydrogenase (Abstract and elsewhere). The formal definition of a dehydrogenase is an enzyme that catalyzes the removal of hydrogen atoms from a substrate. The authors have not demonstrated this for EtfD. The dehydrogenase in the system is the ACAD. EtfD and EtfAB appear to be electron transfer components that enable the ACAD to turnover. This is clear in Figure 5, but is especially confusing in the Abstract.

We thank the reviewer for raising this point. We corrected the manuscript accordingly.

2. In stating that cholesterol degradation yields acetyl-CoA, propionyl-CoA, succinyl-CoA and pyruvate, the authors cite a review article (ll. 40-41). They should cite the original literature.

The references of the original literature were added to the manuscript.

3. Considering that cholesterol catabolism is featured in the Introduction and that this catabolism depends on ACADs, the authors should test whether the etfD and etfAB mutants grow on cholesterol. More specifically, ACADs are involved in degrading both the cholesterol side chain and some of the rings (Thomas and Sampson. 2013. Biochemistry 52, 2895-2904. DOI: 10.1021/bi4002979; Gadbery et al. 2020. Biochemistry 59, 1113-1123. doi: 10.1021/acs.biochem.0c00005). Evaluating the involvement of the Etf proteins in cholesterol catabolism would broaden the significance of this study. In addition, establishing whether growth on cholesterol is impaired would provide a better context for interpreting the data obtained in the infection model.

The reviewer raises an interesting hypothesis. We have tested this hypothesis and found that both \$\Delta\$ etfD and \$\Delta\$ etfBA cannot grow in media with cholesterol as a single carbon source (please see revised Fig 3, Fig 6 and Supplementary Fig 4). Please see lines 114-115, 120-123 and 172, 177 in results section and lines 213-214, and 225-229 in the discussion section. We thank the reviewer for this suggestion.

4. The conclusions that EtfAB forms a complex and is a flavoprotein should be strengthened (ll. 211-212). Related to this, it is noted that only EtfB co-precipitated with EtfD. Specifically, the authors should produce EtfAB heterologously. The proteins appear to be cytoplasmic, so if one component was produced with a His-tag, it should be quite straightforward to demonstrate that the two proteins form a single, multimeric complex that contains flavin.

We agree with the reviewer that showing EtfBA to be a heterodimer binding FAD would strengthen the manuscript. We have tried to express a EtfA-His tag / EtfB- Flag tag construct using the plasmid pET-26b. So far, we have not been successful with the co-overexpression of the 2 proteins. Determining the physical interactions not only of EtfA and EtfB, but also EtfD-EtfBA and ACADs-EtfBA are part of our future work. Although it is an important point, we hope that the reviewer considers that the optimization needed for these experiments to be successful extends beyond a reasonable amount of time for this revision process and might require a collaboration with a team more experienced in protein overexpression and purification. Nevertheless, we took advantage of the new tool AlphaFold2 (Tunyasuvunakool K, 2021 PMID: 1295309) to generate predictions of EtfB and EtfA structures and used the predicted structures to analyze a possible interaction through the docking software HDOCK server (Yan Y, 2020 PMID: 32269383). The results showed that the interaction of EtfB and EtfA is very likely (predicted ligand rmsd 0.22 Å; as a reference value, strong interactions normally have a ligand rmsd <2 Å, Gao Y, 2007 PMID: 17803215).

5. In Figure 5b, the authors should indicate that the system requires a terminal electron acceptor (i.e., the equivalent of ubiquinone that serves this function for the human homologs). Depicting a generic requirement would suffice.

Figure 5b was changed according to the reviewer's suggestions.

REVIEWERS' COMMENTS

Reviewer #1 (Remarks to the Author):

The authors have attempted to address the concerns I highlighted in my review. However, they acknowledge being unable to complete the required experiments in the time available for resubmission. The authors show histopathological data which suggest that EtfBA are required in vivo.

Reviewer #2 (Remarks to the Author):

The authors have adequately responded to my questions and requests for additional information.

Just one detail: reference to new Fig.1d is missing in text, as well as statistical comparisons of shown data. The associated comment l.89 should be corrected accordingly, e.g. 'did not reduce growth to the same extent'.

Reviewer #3 (Remarks to the Author):

The authors have addressed my comments. Specifically, I agree that further characterization of EtfAB is not necessary for this publication. However, I think it would be useful to explicitly mention the inability of the *etfD* mutant to grow on cholesterol in the Abstract.

We thank the editor and the reviewers for their positive feedback and helpful comments. We have addressed the specific points raised in their reviews. Below, please find the reviewers' comments, followed by our responses.

Reviewer #1 (Remarks to the Author):

The authors have attempted to address the concerns I highlighted in my review. However, they acknowledge being unable to complete the required experiments in the time available for resubmission. The authors show histopathological data which suggest that EtfBA are required in vivo.

Thank you for understanding that the EtfBA mouse study will require significant additional time.

Reviewer #2 (Remarks to the Author):

The authors have adequately responded to my questions and requests for additional information.

Just one detail: reference to new Fig.1d is missing in text, as well as statistical comparisons of shown data. The associated comment l.89 should be corrected accordingly, e.g. 'did not reduce growth to the same extent'.

Thank you for making us aware of these issues, they have been addressed in the revised manuscript.

Reviewer #3 (Remarks to the Author):

The authors have addressed my comments. Specifically, I agree that further characterization of EtfAB is not necessary for this publication. However, I think it would be useful to explicitly mention the inability of the *etfD* mutant to grow on cholesterol in the Abstract.

We have included that the *etfD* mutant cannot grow on cholesterol in the abstract.